# Fusing Echocardiography Images and Medical Records for Continuous Patient Stratification

**Nathan Painchaud**[1,2]                    NATHAN.PAINCHAUD@USHERBROOKE.CA
**Pierre-Yves Courand**[1,3,4]               PIERRE-YVES.COURAND@CHU-LYON.FR
**Pierre-Marc Jodoin**[2]                    PIERRE-MARC.JODOIN@USHERBROOKE.CA
**Nicolas Duchateau**[1,5]                   NICOLAS.DUCHATEAU@CREATIS.INSA-LYON.FR
**Olivier Bernard**[1]                       OLIVIER.BERNARD@INSA-LYON.FR

[1] *Univ Lyon, INSA-Lyon, Université Claude Bernard Lyon 1, UJM-Saint Etienne, CNRS, Inserm, CREATIS UMR 5220, U1294, F-69621, Lyon, France*

[2] *Department of Computer Science, University of Sherbrooke, Sherbrooke, QC, Canada*

[3] *Cardiology Dept., Hôpital Croix-Rousse, Hospices Civils de Lyon, Lyon, France*

[4] *Cardiology Dept., Hôpital Lyon Sud, Hospices Civils de Lyon, Lyon, France*

[5] *Institut Universitaire de France (IUF)*

**Editors:** Accepted for publication at MIDL 2024

## Abstract

Deep learning now enables automatic and robust extraction of cardiac function descriptors from echocardiographic sequences, such as ejection fraction or strain. These descriptors provide fine-grained information that physicians consider, in conjunction with global variables from the clinical record, to assess patients' condition. Drawing on novel transformer models applied to tabular data (e.g. variables from electronic health records), we propose a method that considers descriptors extracted from medical records and echocardiograms to learn a representation of hypertension, a difficult-to-characterize and highly prevalent cardiovascular pathology. Our method first embeds each descriptor separately using modality-specific approaches. These embeddings are fed as tokens to a transformer encoder, which combines them into a unified representation of the patient to predict a clinical rating. This task is formulated as an ordinal classification to enforce a pathological continuum in the representation space. We observe trends along this continuum for a cohort of 239 hypertensive patients to describe the gradual effects of hypertension on cardiac function descriptors. Our analysis shows that i) pretrained weights from a foundation model allow to reach good performance (83% accuracy) even with limited data ($< 200$ training samples), ii) trends across the population are reproducible between trainings, and iii) for descriptors known to interact with hypertension, patterns are consistent with prior physiological knowledge.

**Keywords:** Multimodal, contrastive learning, transformer, foundation model, cardiac ultrasound, health records, hypertension

## 1. Introduction

When assessing patients, the typical clinical workflow is to integrate complementary data from various sources such as medical images and Electronic Health Records (EHRs) (Zhou et al., 2023; Hager et al., 2023) into an overall picture of the patient's status. Such a workflow is especially relevant for conditions with a complex pathophysiology, like hypertension (HT) (Mancia et al., 2023). However, the fusion of heterogeneous data makes it challenging to properly assess HT using machine and deep learning methods.

In this paper, a summary of (Painchaud et al., 2024), we use transformers to combine EHR data and clinical parameters extracted from 2D+time echocardiographic sequences to characterize HT. We base our method on the tabular representation of EHR data, with a branch to integrate image data. We also formulate a supervised finetuning objective to predict a position along a pathological continuum given only a few target classes.

## 2. Method

Figure 1 illustrates our pipeline. Echocardiographic sequences are segmented with a state-of-the-art model (1a) (Ling et al., 2023) and descriptors are extracted from these segmentations (1b). In parallel, health records are structured into categorical and scalar descriptors (2). These descriptors are embedded using modality-specific methods for time-series w.r.t. the cardiac cycle (3a) (Pellegrini et al., 2023) and tabular data (3b) (Gorishniy et al., 2021). The embeddings are fed to a transformer encoder (4a) that uses pretrained weights from the XTab foundation model (Zhu et al., 2023). From the encoder's latent space, a classification head predicts the patient stratification (5), with the constraint that the probability distribution over the classes must be unimodal to model the order of the classes (Beckham and Pal, 2017). The latent space can then be visualized and analyzed w.r.t. the predicted stratification (6) (cf. Figure 2).

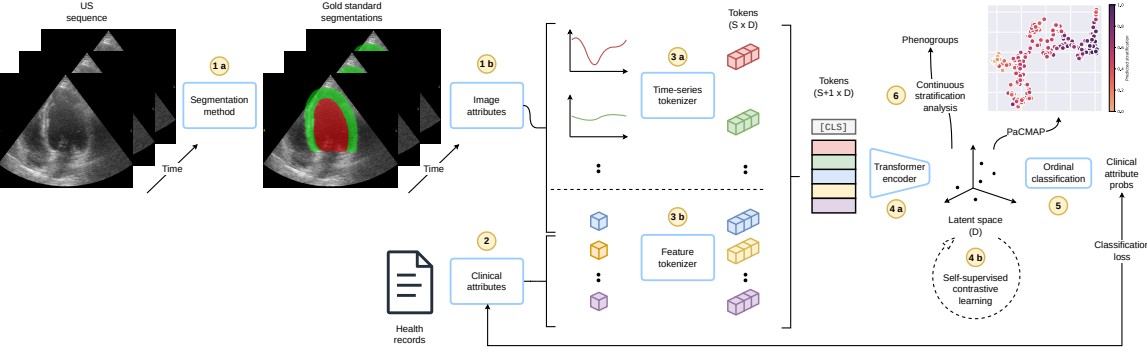

Figure 1: Schematic representation of our multimodal fusion pipeline, from ultrasound images and EHRs to patient stratification.

## 3. Results and Discussion

We tested our method on a private dataset of echocardiograms — Apical 4 Chamber (A4C) and Apical 2 Chamber (A2C) views — and EHR data — 53 numerical and categorical variables — from 239 hypertensive patients. The target HT severity descriptor corresponds to three degrees of hypertension assessed by a cardiologist.

We performed ablation studies to evaluate configurations of pretrained weights and input data. Table 1(a) shows that using the XTab foundation model's weights drastically improves performance (around 20%) over random initialization and contrastive pretraining (Bahri et al., 2022; Onishi and Meguro, 2023) on our private dataset. Table 1(b) also highlights that our method scales well given more input descriptors, with accuracy plateauing when new data is not discriminative. This sets our method apart from similar analyses that filter redundant or noisy features beforehand (Zheng et al., 2020).

Table 1: Ablation study of the transformer encoder pipeline. Each result corresponds to the mean ± standard deviation over 10 trainings with the same configuration.

(*a*) Transformer encoder's weight initialization

| Weights init. | random | pretrained | xtab |
|---|---|---|---|
| Accuracy (%) | 63.5 ± 6.2 | 56.9 ± 7.2 | **83.3 ± 2.8** |

(*b*) Descriptors provided as input

| # input desc. | 27 | 64 | 78 |
|---|---|---|---|
| Accuracy (%) | 74.4 ± 3.8 | **83.5 ± 4.8** | 83.3 ± 2.8 |

We also evaluate the clinical relevance of the representation. Figure 2(*a*) depicts the latent space, where the patients are monotonically distributed w.r.t. HT severity. Figure 2(*b*) plots average global longitudinal strain (GLS) curves, a standard descriptor of the heart's contraction over time (Amzulescu et al., 2019), across bins of patients with similar predicted stratification. Differences between curves, like 1 differing from 2 by a steeper slope between 0.6–0.8 on the x-axis, represent subtle alterations of cardiac deformation that future clinical studies could investigate as possible early biomarkers of HT.

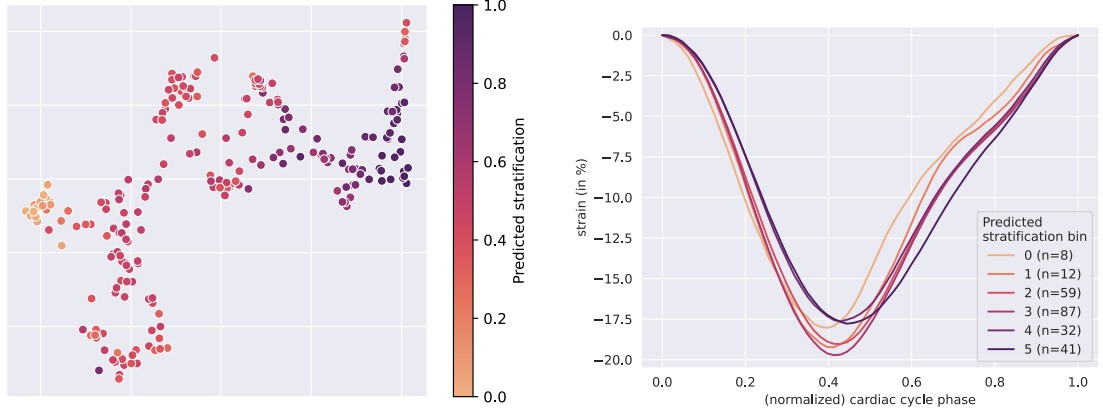

(*a*) 2D embedding of the 192D latent space, using PacMAP (Wang et al., 2021). Each point is a patient, colored w.r.t. to the predicted stratification.

(*b*) Average curves of Global Longitudinal Strain (GLS), describing cardiac deformation, over bins of patients grouped by predicted stratification.

Figure 2: Visualization and analysis of the HT representation learned by the model.

## 4. Conclusion

We proposed a framework for fusing tabular data and 2D+time echocardiograms to learn a stratification of patients, given limited data with categorical labels. The framework is designed for difficult-to-characterize pathologies since it combines complementary data from multiple sources and can pinpoint the patients along an interpretable pathological continuum. We showed that our continuous stratification allows insights into how hypertension can subtly affect clinically-relevant cardiac function descriptors. Our pipeline could help clinicians study early alterations of cardiac biomarkers, contributing to improve our understanding of complex pathologies.

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
