# OpenReview forum: "Fusing Echocardiography Images and Medical Records for Continuous Patient Stratification"
_MIDL.io/2024/Short_Papers — MIDL 2024 Short Papers_

### Official Review · Reviewer_DMHq · 2024-04-24

**Confidence:** 4
**Final Rating:** 5

**Review:**

The short paper, a summary of an arXiv paper, studies hypertension (HT) based on echocardiography and EHRs. echocardiography data is segmented automatically and features are extracted. cardiac time-series are then embedded, while EHRs are also embedded. These embeddings are fed to a XTab model for fusion, and the latent space is used for visualisation using dimensionality reduction, as well as classification into three pre-determined HT groups.

Strengths
- clinical relevance of the task
- good example of how to combine multimodal data
- suprising results on pretrained vs xtab initialisation
- state of the art algorithms used throughout

Weaknesses
- contribution of echo alone vs EHRs alone vs combination not clear, what is the gain?
- no baseline provided to compare accuracy of 3-class HT classification (e.g. XGB on image or EHR features, or embeddings thereof)
- additional (or replacement) latent space figure, with points coloured by HT class, should be shown

---

### Decision · Program_Chairs · 2024-04-26

Accept